# Low-dose ionizing radiation generates a hormetic response to modify lipid metabolism in *Chlorella sorokiniana*
Marina Stanić [1], Mima Jevtović[1,2], Snežana Kovačević[1], Milena Dimitrijević[1], Jelena Danilović Luković[1,3], Owen A. McIntosh[4], Bernd Zechmann [5], Alessandro Marco Lizzul[6], Ivan Spasojević [1] ✉ & Jon K. Pittman [4] ✉

Algal biomass is a viable source of chemicals and metabolites for various energy, nutritional, medicinal and agricultural uses. While stresses have commonly been used to induce metabolite accumulation in microalgae in attempts to enhance high-value product yields, this is often very detrimental to growth. Therefore, understanding how to modify metabolism without deleterious consequences is highly beneficial. We demonstrate that low-doses (1–5 Gy) of ionizing radiation in the X-ray range induces a non-toxic, hormetic response in microalgae to promote metabolic activation. We identify specific radiation exposure parameters that give reproducible metabolic responses in *Chlorella sorokiniana* caused by transcriptional changes. This includes up-regulation of >30 lipid metabolism genes, such as genes encoding an acetyl-CoA carboxylase subunit, phosphatidic acid phosphatase, lysophosphatidic acid acyltransferase, and diacylglycerol acyltransferase. The outcome is an increased lipid yield in stationary phase cultures by 25% in just 24 hours, without any negative effects on cell viability or biomass.

Realising sustainable production and consumption goals requires a shift away from use of non-renewable materials and production processes that cause environmental degradation and have a negative carbon balance, and a transition towards the utilization and exploitation of renewable biological materials[1,2]. Microalgae represents one such renewable biomass source with multiple industrial applications due to its versatility to produce a wide variety of biopolymers, chemicals, and metabolites, and flexibility to be cultivated under conditions that require limited inputs, in contrast to intensive agriculture[3–5]. As such, microalgae biomass is an alternative sustainable feedstock for animal feed, food ingredients and biofuels production, particularly in relation to high content of lipids[6,7]. Enhancing lipid yield, especially triacylglycerol (TAG) yield, is therefore essential for the techno-economic performance of many microalgae applications[8]. Microalgae-derived TAGs can be used in the production of cosmetics, can be chemically converted into a biodiesel, or used as an alternative to vegetable oils, such as the currently unsustainable palm oil[5]. While approaches including the development of genetically engineered strains or the modification of strains by random mutagenesis are being considered, there are still gaps in capability. Moreover, these approaches have challenges including commercial

acceptability, the requirement for laborious screening and the risk of reversion of mutations to a wild-type state[5,9,10]. A more common approach uses external stressors, such as cultivation under nutrient limitation, to shift metabolism of natural strains of microalgae into a lipid production mode. However, this is to the detriment of biomass productivity, cultivation process complexity, time and cost[11]. High doses of electromagnetic radiation have commonly been used as a means to mutate microalgae, most often by exposure to ultraviolet radiation[12], but also in some cases by using ionizing radiation such as gamma-rays[13,14]. High doses of ionizing radiation will cause direct DNA damage and will typically induce the production of reactive oxygen species (ROS) that may result in irreversible damage and cell death. However, few studies have examined non-toxic and non-mutagenic responses of microalgae to low doses of ionizing radiation in an attempt to induce increased production of biomass and metabolites. Radiation hormesis is a process whereby an organism displays a beneficial response following exposure to low doses of ionizing radiation, which is otherwise toxic at higher doses, but this phenomenon is poorly understood in plants and algae[15,16]. In plants, hormetic doses of ionizing radiation have been found to enhance growth rate and provide other stimulatory effects such as

---

[1]University of Belgrade—Institute for Multidisciplinary Research, Life Sciences Department, Belgrade, Serbia. [2]Innovative Centre of the Faculty of Chemistry, University of Belgrade, Belgrade, Serbia. [3]Institute for Application of Nuclear Energy—INEP, University of Belgrade, Belgrade, Serbia. [4]Department of Earth and Environmental Sciences, School of Natural Sciences, The University of Manchester, Manchester, UK. [5]Center for Microscopy and Imaging, Baylor University, Waco, TX, USA. [6]Varicon Aqua Solutions Ltd., Worcester, UK. ✉e-mail: redoxsci@gmail.com; jon.pittman@manchester.ac.uk

enhanced antioxidant characteristics[16]. The exact mechanisms of these responses are unclear but a key component in transducing the irradiation response is likely to be via the formation of ROS by water radiolysis[17]. ROS can then act as signaling molecules to initiate redox regulation of downstream processes such as cell division and other biochemical and physiological responses, via transcriptional changes[18,19]. However, the physiological consequences and molecular responses to ionizing radiation hormesis in microalgae are unclear, particularly with regard to metabolic modification. The green microalga *Chlorella sorokiniana* is a widely studied species that is commercially relevant in terms of its resilient and rapid growth[20], high productivity of metabolites including storage lipids for various applications[21,22], and suitability for genomic-scale gene expression analysis[23,24]. Here we report a hormesis-induced re-engineering of lipid metabolism in *C. sorokiniana*, demonstrating a rapid increase in lipid yield by selection of beneficial doses and dose rates of ionizing radiation in the X range of the electromagnetic spectrum at specific time points during microalgal cultivation. Furthermore, we show that biomass yield and cell viability are not affected by the treatment and that the response is highly reproducible, demonstrating that the trait is not due to radiation-induced mutation. Finally, we identify the transcriptional targets of the X-ray activation pathway, revealing that lipid metabolism is modified following low-dose irradiation by induction of genes encoding core components of fatty acid synthesis and TAG synthesis.

## Results

### Low-dose irradiation during exponential phase increases cell growth rate and biomass

We developed and optimized a method for effective low dose X-ray irradiation of microalgae to significantly enhance metabolite yield through three steps, using *C. sorokiniana* strain CCAP 211/8 K. Step 1 screened radiation treatments applied during early exponential phase of growth. Step 2 evaluated early stationary phase irradiation using a narrower range of doses and dose rates. Finally, Step 3 examined an even narrower range of doses and dose rates applied during early stationary phase in more detail, including an evaluation of rapid responses after just 24 h following irradiation (Fig. 1a).

　　Examination of irradiation during early exponential phase at 3 days into cell growth (Step 1) showed that under certain irradiation conditions lower doses (1–5 Gy) of irradiation promoted growth over 30 days at specific dose rates, tested at 0.05, 0.25 and 0.5 Gy/min, whereas higher doses (10 and 20 Gy) had the opposite effect (Fig. 1b, c). For example, 1, 2 and 5 Gy irradiation treatments significantly increased (1 Gy, $p = 0.048$; 2 Gy, $p = 0.031$; 5 Gy, $p = 0.048$) growth rate at 0.25 Gy/min, by 43%, 44% and 34%, respectively. Likewise, 2 and 5 Gy irradiation treatments at 0.05 Gy/min significantly increased ($p = 0.048$) growth rate. In contrast, 20 Gy irradiation at 0.5 Gy/min was very inhibitory to *C. sorokiniana* growth (Fig. 1b). Some of the cultures with more rapid growth provided increased dry weight biomass in the stationary phase, specifically at day-30 with 2 Gy at 0.05 Gy/min, and day-25 with 1 Gy at 0.25 Gy/min (Fig. 1d). Although the exponential phase X-ray exposure did not significantly increase lipid yield, as determined by Nile Red-positive neutral lipid staining (Fig. 1e), volumetric increase of lipid content by day-30 was observed, which was significantly increased ($p = 0.048$) in the 1 Gy at 0.25 Gy/min and 5 Gy at 0.5 Gy/min treatments (Supplementary Fig. 1a). It was also notable that high irradiation exposure (10 Gy at 0.25 Gy/min and 20 Gy at 0.5 Gy/min) significantly inhibited ($p = 0.009$) both lipid yield (Fig. 1e) and volumetric lipid content (Supplementary Fig. 1a). There was also no deleterious change in cellular content of chlorophyll a, chlorophyll b or total carotenoid pigments following low-dose irradiation (Supplementary Fig. 2).

### Low-dose irradiation during stationary phase rapidly increases lipid yield without inhibiting biomass or cell viability

Step 2 investigated low-dose X-ray irradiation applied at early stationary phase cells (day-20 of growth) using a selected range of doses and dose rates (2 Gy at 0.05 Gy/min; 1, 2 and 5 Gy at 0.25 Gy/min; 2 and 5 Gy at

0.5 Gy/min) that were found to avoid growth inhibition when applied at exponential phase (Fig. 1a). No negative effects on cell density and biomass were observed 10 days after irradiation (Fig. 1f, g). Likewise, there was no alteration in the content of chlorophyll and carotenoid pigments (Supplementary Fig. 3a). In contrast, lipid yield at 10 days after irradiation was significantly increased ($p < 0.05$) in comparison to the control cells for all of the irradiation treatments tested (Fig. 1h). This included a 32% and 29% increase in lipid for the cells exposed to 1 Gy at 0.25 Gy/min ($p < 0.0001$) and 5 Gy at 0.5 Gy/min ($p = 0.0001$), respectively. As expected, these treatment conditions also gave high volumetric lipid content in these stationary phase cultures (Supplementary Fig. 1b). For some of the treatments there was also a significant increase ($p = 0.037$) in cell volume (Fig. 1i), and a significant reduction ($p = 0.012$) in starch yield (Fig. 1j).

　　As stationary phase irradiation was successful at enhancing microalgae lipids, Step 3 examined whether a strong increase in lipid yield could be reached in just 24 h following early stationary phase irradiation (Fig. 1a), since time efficiency is important for successful commercial application of microalgae[25]. All tested doses (1 Gy at 0.25 Gy/min, $p = 0.002$; 2 Gy at 0.25 Gy/min, $p = 0.005$; and 5 Gy at 0.5 Gy/min, $p = 0.001$) resulted in significant lipid yield increases of ~25%, as determined from quantification of total extracted lipids (Fig. 2a and Supplementary Fig. 1c), without reducing biomass (Fig. 2b). Cell viability was also unaffected, as determined by Evans Blue staining to distinguish dead or damaged cells (Fig. 2c), and other physiological markers such as the concentrations of photosynthetic pigments remained unchanged (Supplementary Fig. 3b).

　　Lipid characteristics were examined in more detail. The fatty acid profile from extracted *C. sorokiniana* lipids, which was rich in C16:0, C16:1, C18:2 and C18:3 fatty acids, was not negatively affected by the irradiation treatments (Fig. 2d). In fact, for the cells exposed to 1 Gy at 0.25 Gy/min the amounts of many of the fatty acids including C15:1, C16:0, C16:2, C16:3 and C17:0 were significantly increased ($p = 0.034$) compared to the control treatment. The calculated cetane number, as a determinant of the ignition quality of a biodiesel fuel, was unaffected by irradiation (Fig. 2d); therefore, lipids remained compatible with biodiesel production[26]. The fatty acid profile also indicates that the TAGs remain ideal for nutritional applications due to the high abundance of essential polyunsaturated fatty acids including linoleic acid (C18:2), an omega-6 fatty acid, and α-linolenic acid (C18:3), an omega-3 fatty acid. In fact, the proportion of α-linolenic acid within the extracted lipid was significantly increased ($p = 0.034$) by the 1 Gy at 0.25 Gy/min irradiation treatment from 9.31 mg/g to 22.89 mg/g (Fig. 2d). The irradiation-induced enhancement of cytosolic lipid droplets, which is where the TAGs mainly accumulate, was confirmed by transmission electron microscopy (TEM) (Fig. 2e). This analysis showed both increased number and size of lipid droplets, 24 h after irradiation (Fig. 2f), alongside an increase in cell volume (Fig. 2g), which is a previously observed relationship when lipid droplets increase in number[27]. In contrast, the number and size of starch granules, and the amount of starch relative to biomass, were all lowered by irradiation treatment (Fig. 2h, i), implying carbon mobilization from starch to lipids, as observed previously in response to other stressors[28]. However, there is some increase in carbohydrate content in *C. sorokiniana* cells following irradiation in the form of cell wall polymers. This was determined in previous experiments where we showed that 1–5 Gy of X-ray treatment was able to give rise to a thicker cell wall and increased cell wall yield[29]. The low-dose irradiation-induced increase in lipid yield and the maintenance of biomass yield were not transient responses, but were still observed 10 days after irradiation with 2 and 5 Gy (Supplementary Fig. 4).

### Rapid transcriptional up-regulation of lipid metabolism in early stationary phase irradiated cells

Ionizing radiation can induce stress through radiolytic production of ROS that may result in a metabolic response or in irreversible damage and cell death[30]. Here, low-dose irradiation of stationary phase cells did not inhibit cell viability (Fig. 2c), but instead induced metabolic activation as determined by the increase in lipid metabolism (Fig. 2a) and the transcriptional enhancement of multiple metabolic pathways (Fig. 3). These metabolic

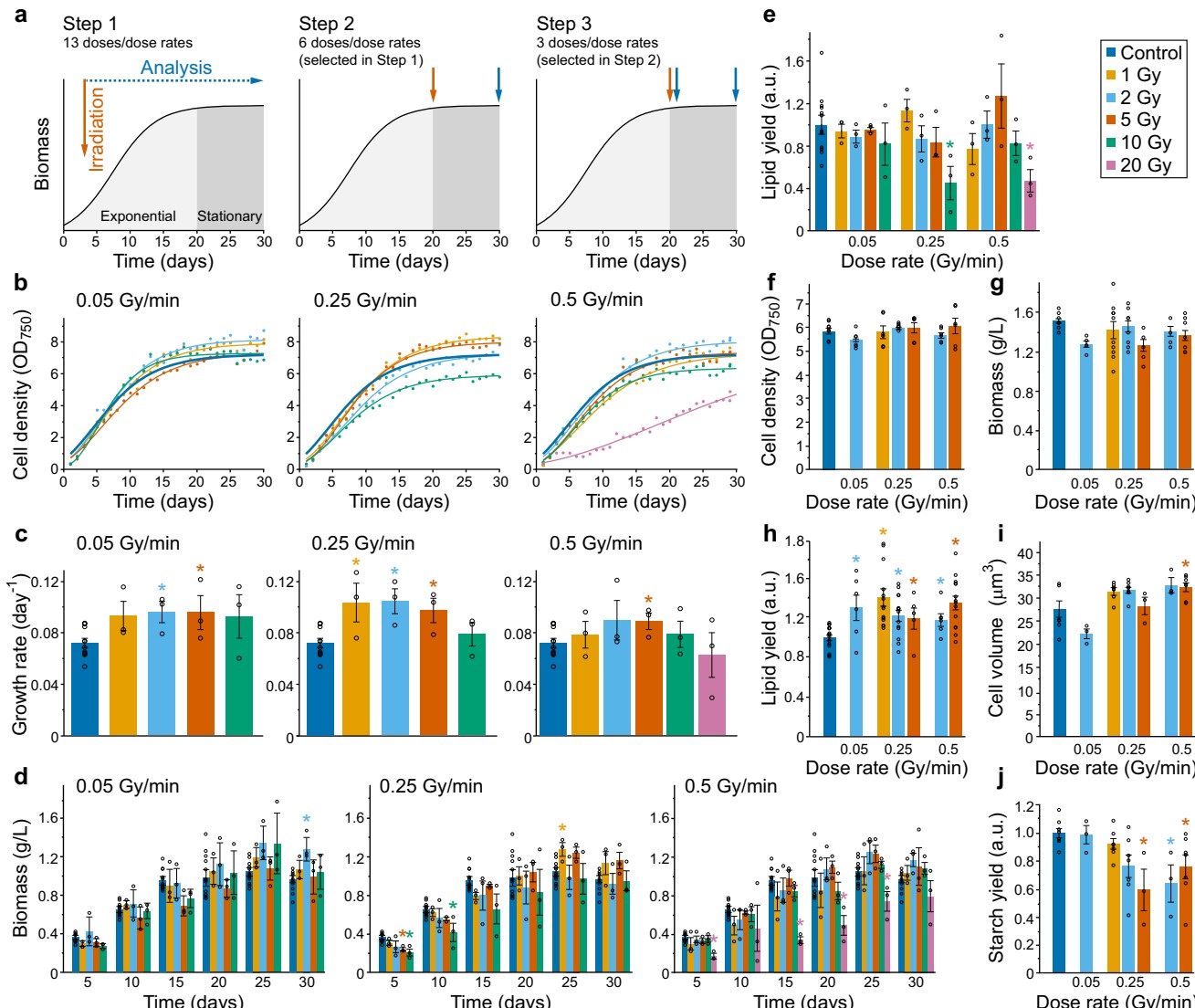

**Fig. 1 | Optimization of the irradiation protocol to maximize lipid yield in** *Chlorella sorokiniana* **cultures. a** Experimental design of the 3-step optimization. Time points of irradiation and analysis are marked with arrows on the schematic culture growth curve. Cultures were irradiated with different doses (Gy) and dose rates (Gy/min) of X-radiation after 3 or 20 days (early exponential or early stationary phase). At each step the selection of conditions were narrowed to those that delivered positive changes. **b**–**e** Effects of irradiation applied in early exponential phase measured over 30 days after irradiation (Step 1). The number of biologically independent experiments were $n = 12$ for controls and $n = 3$ for treatments. **b** Cell density by optical density at 750 nm ($OD_{750}$). Sigmoidal data fitting (all $R^2 > 0.95$) is shown. **c** Growth rates during exponential phase. **d** Dry weight biomass. **e** Lipid yield in

biomass at day-30 after irradiation determined by Nile Red fluorescence (a.u., arbitrary units). **f**–**j** Effects of irradiation applied in early stationary phase measured 10 days later (Step 2). **f** Cell density. **g** Biomass. The number of biologically independent experiments were $n = 8$ for controls and $n = 6$–10 for treatments. **h** Lipid yield in biomass. The number of biologically independent experiments were $n = 16$ for controls and $n = 5$–14 for treatments. **i** Cell volume. **j** Relative starch yield in biomass. The number of biologically independent experiments were $n = 7$ for controls and $n = 3$–7 for treatments. All data are means ± standard error. Significant difference compared to control (non-irradiated) treatments was evaluated using non-parametric two-tailed Mann–Whitney U tests and is indicated at $p < 0.05$ (*).

responses are proposed to be induced via redox signaling[18,19], as illustrated in Supplementary Fig. 5. ROS including hydroxyl radicals ($HO^{\bullet}$), hydrogen peroxide ($H_2O_2$) and superoxide radical anions ($O_2^{\bullet-}$) can be generated by radiolysis, and have varying diffusion radii and ability to reach different targets across a cell[17,31]. For $HO^{\bullet}$ the diffusion radius is only ~3 nm compared to >1 μm for $H_2O_2$. While $HO^{\bullet}$ is extremely short-lived and only induces damage to biomolecules (such as carbohydrates, lipids and DNA) near the site of production, $O_2^{\bullet-}$ is converted by intracellular superoxide dismutases into $H_2O_2$, which has a longer half-life time (~5 ms) and can pass through membranes (Supplementary Fig. 5). $H_2O_2$ signaling controls transcription factors and the activity of various metabolic proteins through oxidative modifications of thiol switches[32]. Transcription factors and proteins that are redox regulated in plants[18,33,34], and are found to be expressed in

*C. sorokiniana*, include members of the AP2/ERF, bZIP and MYB classes of transcription factors, and signaling proteins including MAP kinases, cyclins, and cyclin-dependent kinases (Supplementary Fig. 5). In contrast to the signaling role of low-level ROS, an excess of these species results in cell damage and may explain the cytotoxic effects of higher doses of radiation observed here (Fig. 1b–d). Pertinent to this, it was demonstrated by EPR spectroscopy that the dose with hormetic effects (5 Gy at 0.5 Gy/min) did not obviously alter the redox milieu as compared to the control treatment (Supplementary Fig. 6). Irradiation resulted in the reduction of a $Cu^{2+}$ signal (reduced to 'EPR silent' $Cu^{1+}$). Otherwise, no redox alterations could be observed.

The wide range of the metabolic activation was characterized by extensive transcriptional induction (266 genes significantly increased with

**Fig. 2 | Increased lipid yield following low-dose irradiation in early stationary phase cultures.**
**a–h** Effects of irradiation applied in early stationary phase measured 1 day later (Step 3). **a** Lipid content in biomass determined by lipid extraction and gravimetry. The number of biologically independent experiments was $n = 8$. **b** Biomass. The number of biologically independent experiments was $n = 4$. **c** Cell viability. The number of biologically independent experiments was $n = 8$. **d** Fatty acids profile with fractions of saturated (SFA), mono-unsaturated (MUFA), and poly-unsaturated fatty acids (PUFA), and cetane values of total fatty acids (shown inset). The number of biologically independent experiments was $n = 3$ for control and $n = 3$–4 for treatments. **e** Representative TEM micrographs of control and irradiated cells (N, nucleus; C, chloroplast; LD, lipid droplets; S, starch granules). **f** Number and proportional cross-section area of lipid droplets. The number of biologically independent experiments was $n = 3$; the number of analyzed cells was $n > 25$. **g** Cell volume. **h** Number and proportional cross-section area of starch granules. **i** Total starch content in biomass. The number of biologically independent experiments was $n = 7$–8. All data are means ± standard error. Significant difference compared to control (non-irradiated) treatments was evaluated using non-parametric two-tailed Mann–Whitney U tests (one-tailed in (**d**)) and is indicated at $p < 0.05$ (*).

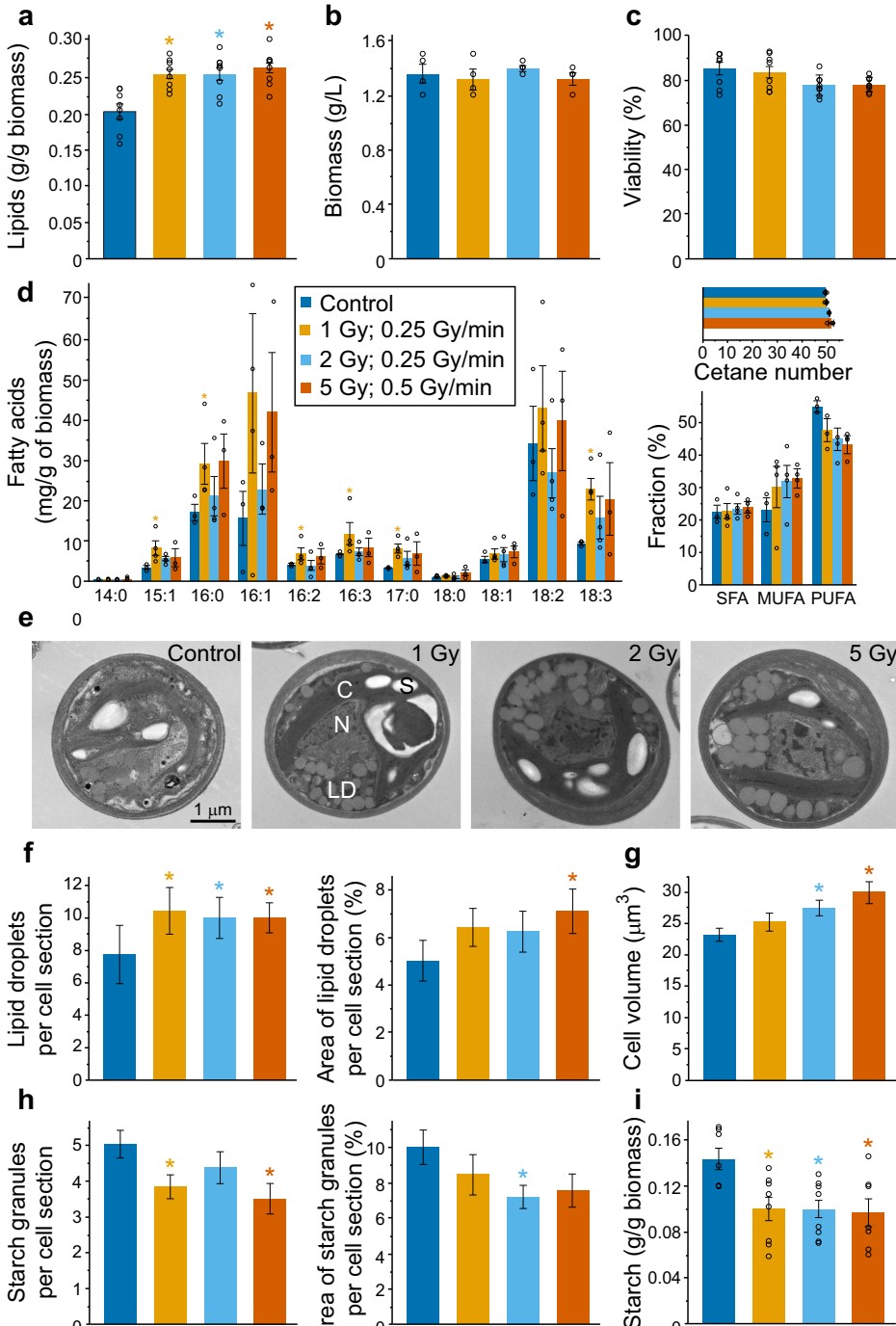

log2 fold-change (FC) > 0.5, false discovery rate (FDR) < 0.05; of which 136 genes had a log2 FC > 1) and some down-regulation (144 genes significantly decreased with log2 FC < −0.5, FDR < 0.05; of which 81 genes had a log2 FC < −1) in response to 5 Gy at 0.5 Gy/min irradiation in comparison to the control (Fig. 3a). This included a large number of up-regulated transcripts related to key metabolic and homeostatic functions, such as amino acid metabolism, photosynthesis, DNA replication and repair, redox homeostasis, and RNA processing and modification (Fig. 3b). The largest number of up-regulated transcripts were annotated to lipid metabolism (39 genes) (Fig. 3b and Supplementary Data 1) and include those encoding critical enzyme steps in fatty acid synthesis and TAG biosynthesis (Fig. 3d). The up-regulated transcripts were mapped to various lipid metabolism pathways, including the chloroplastic fatty acid synthase pathway, fatty acid elongation, and glycer-olipid metabolism (Supplementary Fig. 7). The up-regulated enzymes of the

fatty acid synthase pathway (Fig. 3d) included the biotin carboxylase complex of acetyl-CoA carboxylase (ACCase) and four different isoforms of long-chain acyl-CoA synthetase (LACS), which play essential roles in lipid accu-mulation. Significant ($p < 0.05$) irradiation-induced increase in expression for these five fatty acid metabolism genes was confirmed by quantitative real-time PCR (qPCR) and showed a 2.8-fold increase in ACCase expression ($p = 0.011$), and a 2.4-fold increase in expression for one of the LACS isoforms ($p = 0.021$) (Fig. 3c and Supplementary Fig. 8). The ACCase catalyzes the first key step of fatty acid biosynthesis and previously has been found to be an up-regulated enzyme in microalgae with enhanced lipid accumulation[35,36]. LACS isoforms have also been shown to be important in lipid homeostasis in response to stress conditions[37].

Up-regulated transcripts that encode TAG biosynthesis enzymes were mapped to the glycerolipid biosynthesis (Kennedy) pathway (Fig. 3d) and

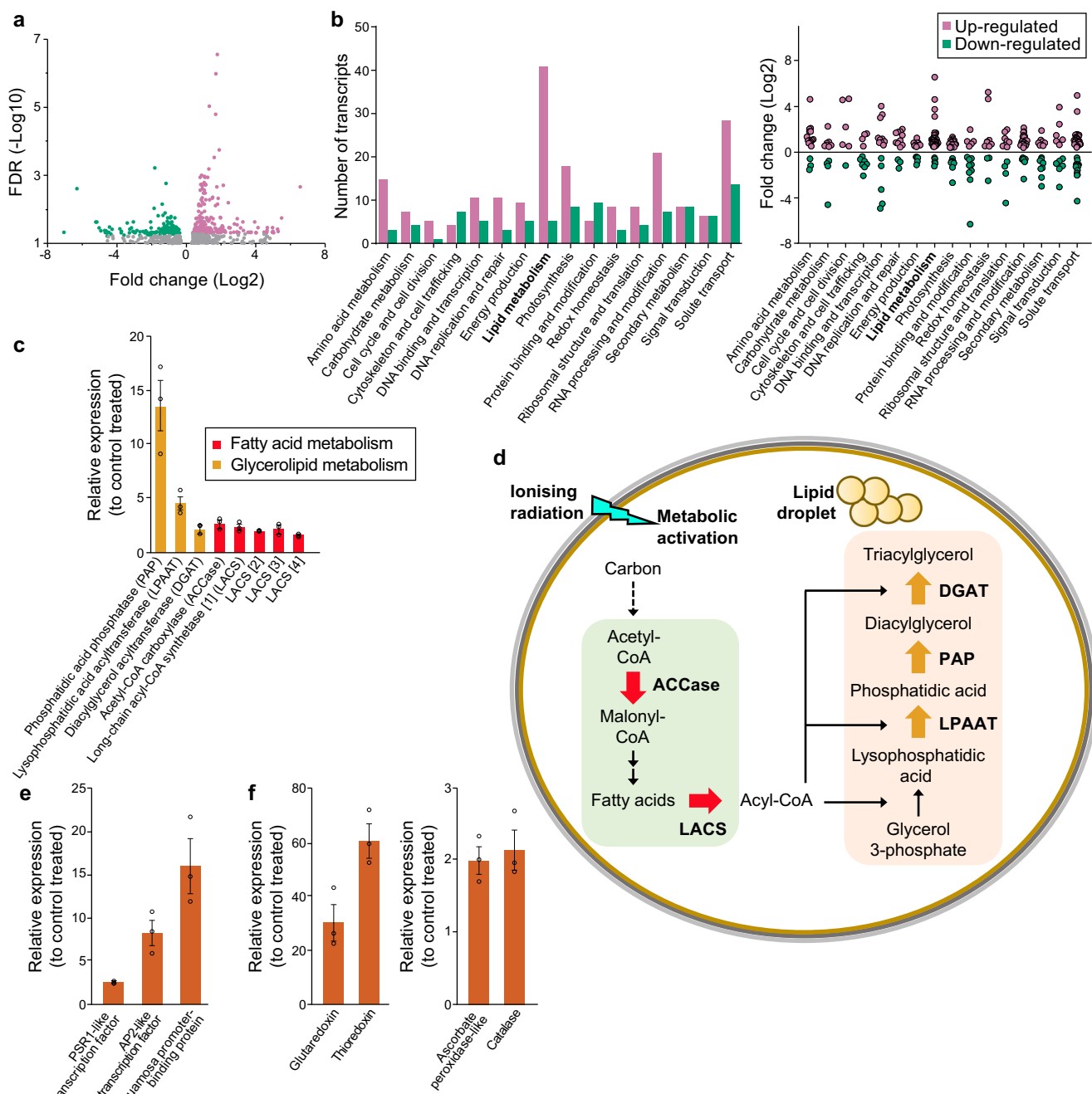

**Fig. 3 | Irradiation induced changes in gene expression in early stationary phase.**
**a** 'Volcano' plot of differentially expressed genes in cells 1 day after irradiation with
5 Gy at 0.5 Gy/min relative to control (non-irradiated) cells. Highlighted genes were
significantly up-regulated (266 genes) or down-regulated (144 genes) following
irradiation. **b** The number of genes in different gene classes with significant (false
discovery rate (FDR) < 0.05) up- or down-regulation of expression and the fold-
change of their expression. **c** Relative expression of selected lipid metabolism genes
in response to irradiation as determined by qPCR and shown relative to the control
(non-irradiated) expression. Numbers in square brackets represent isoforms. **d** The

mapping of the enzymes associated with the key lipid metabolism genes that were
up-regulated in response to ionizing radiation. (**e**) Relative expression of three
transcription factor genes as determined by qPCR and shown relative to the control
(non-irradiated) expression **f** Relative expression of four reactive oxygen species
response genes as determined by qPCR and shown relative to the control (non-
irradiated) expression. All data (in **c**, **e**, **f**) are means ± standard error and the number
of biologically independent experiments was $n = 3$. The differences compared to
control treatments are all significant at $p < 0.05$ as evaluated using non-parametric
two-tailed Mann–Whitney U tests.

the significant ($p < 0.05$) irradiation-induced expression increase for three of
the genes was validated by qPCR (Fig. 3c and Supplementary Fig. 8). They
include an isoform of lysophosphatidic acid acyltransferase (LPAAT) that is
most similar to the *Chlamydomonas reinhardtii* endoplasmic reticulum-
localized *CrLPAAT2*[38], which had a 2.3-fold increase ($p = 0.013$) in
expression compared to control treatment; a phosphatidic acid phosphatase
(PAP) isoform that shows high similarity to *CrPAP1*[39], with a 13.6-fold

increase ($p = 0.03$) in expression; and an isoform of diacylglycerol acyl-
transferase (DGAT) that is most similar to the Type-2 DGAT *CrDGTT4*[40],
which showed a 4.5-fold increase ($p = 0.013$) in expression. Increased
expression of LPAAT, PAP and DGAT enzymes, either individually or in
combination has been shown to be important for gaining increased TAG
accumulation in microalgae[38–41]. Many up-regulated transcripts were also
mapped to the fatty acid degradation pathway (Supplementary Fig. 7).

An increase in expression of lipid catabolism genes, including those for fatty acid degradation (β-oxidation), would seem to be the opposite of what would be expected during an increase in TAG lipids. However, several of the components of fatty acid β-oxidation including acyl-CoA dehydrogenase, 3-ketoacyl-CoA thiolase and hydroxyacyl-CoA dehydrogenase genes have been previously shown to be up-regulated in response to lipid-inducing nitrogen starvation conditions in microalgae such as *Nannochloropsis*[42], possibly to recycle precursors for new TAG synthesis.

At least fifteen genes that encode putative transcription factors (in the 'DNA binding and transcription' functional class) displayed a significant change in expression (FDR < 0.05) in the irradiated cultures (Fig. 3b), with many of these being members of transcription factor families that are known to regulate lipid metabolism in other microalgae[43]. Significant ($p < 0.05$) differential expression of three of the transcription factor genes were further validated by qPCR (Fig. 3e and Supplementary Fig. 8). These include a SQUAMOSA promoter-binding protein (SBP) transcription factor family member that showed a 16.2-fold increase ($p = 0.035$) in expression compared to the non-irradiated control treatment. SBPs from other microalgae, including the *C. reinhardtii* NRR1 protein is a known lipid regulator[44], although it was not possibly to determine with sufficient accuracy whether this SBP gene is a *C. sorokiniana* NRR1 ortholog. However, an ortholog of the *C. reinhardtii* PSR1 (CrPSR1) transcription factor, which is a member of the MYB coiled-coil domain family and another known lipid regulator[45,46], was identified in *C. sorokiniana* (Supplementary Fig. 9). The *C. sorokiniana PSR1* ortholog gave a 2.7-fold increase ($p = 0.005$) in expression in response to irradiation (Fig. 3e). Increased transcript abundance of *CrPSR1* has previously been shown to induce a significant increase in TAG content[45] and cause increased expression of TAG biosynthesis genes including isoforms of PAP, DGAT and glycerol-3-phosphate dehydrogenase[46]. Assuming that the downstream targets of PSR1 are conserved across these green microalga species, this is a strong candidate to be one of the key components of the irradiation-induced TAG accumulation pathway.

The radiolytic production of ROS is proposed to act as a signal to activate downstream cellular responses including metabolic alteration through redox reactions, but it is also essential for the cell to maintain a redox balance through the use of antioxidant systems[18,19,47]. One of the transcription factors that was confirmed by qPCR as showing a 8.4-fold increase ($p = 0.031$) in expression in response to irradiation is a member of the AP2/ERF family (Fig. 3e), some of which are redox regulated and induce expression of antioxidant enzymes[48]. Candidate components of antioxidant defensive systems such as ascorbate peroxidase and catalase, both of which break down $H_2O_2$ into $H_2O$[47,49], were found to be significantly (ascorbate peroxidase, $p = 0.015$; catalase, $p = 0.028$) induced by irradiation by approximately twofold as determined by qPCR (Fig. 3f and Supplementary Fig. 8). Genes encoding glutaredoxin and thioredoxin isoforms were also increased 31.1-fold ($p = 0.041$) and 61.4-fold ($p = 0.009$), respectively (Fig. 3f). These are important redox regulation enzymes within photosynthetic cells and are proposed to play key roles in abiotic stress-induced redox signaling[50].

## Discussion

This study represents a detailed examination of metabolic responses and mechanisms to X-ray radiation hormesis in microalgae. We provide convincing evidence that at low doses, a short duration of X-ray exposure induces transcriptional up-regulation of various biochemical pathways, particularly a modification of lipid metabolism, giving rise to an increased cellular abundance of stored TAG. As well as the up-regulation of critical lipid metabolism enzymes, the increased expression of previously characterized transcriptional regulators, such as PSR1, and redox regulators including glutaredoxin and thioredoxin isoforms allows us to begin to discern a signaling pathway that is likely activated via radiolytic-induced ROS production, in line with current understanding of redox signaling[30,34]. Although *CrPSR1* has previously been found to be induced by phosphorus and nitrogen starvation[45,46], the observation of low-dose irradiation induction seen here further expands the importance of this transcription factor as

a key metabolic regulator that responds to various environmental perturbations. The high conservation of these up-regulated genes across photosynthetic organisms, both land plants and other classes of algae, suggests that there will be conservation of radiation-induced metabolic activation. Therefore, we propose that this treatment strategy could be applied to other organisms both to understand X-ray induced metabolic activation further and to enable the enhanced production of specific metabolites for industrial uses.

Many prior studies have evaluated stress responses of microalgae to understand mechanisms of resilience and adaptation to specific stressors, and to investigate the applications of these responses, such as by the induction high-value chemicals[11,49,51,52]. In line with the definition of 'stress' as a disruptor of microalgal homeostasis and an inducer of metabolic change during the restoration of homeostasis[53], we can also consider the low-dose X-ray radiation as a stressor. Furthermore, this stressor elicits metabolic responses, such as an accumulation of storage lipids that are consistent with other stress responses[11,51]. However, with regard to carbon storage induction, the irradiation treatment seems to be specific to modification of lipid metabolism and failed to show any alteration to starch metabolism including any changes in expression of starch catabolism or starch biosynthesis genes. Nevertheless, low-dose X-ray radiation of 1–5 Gy is a stressor that does not cause significant inhibition in cell growth, in contrast to high lipid inducing stressors like nutrient starvation[44,46].

The work described, provides a template, validated here for *C. sorokiniana*, for a non-invasive method to induce a rapid and reproducible increase in lipid yield at the early stationary phase, which is a common point of microalgal biomass harvesting. Such innovation is required to allow mass cultivation of microalgae for applications including biofuel and nutritional supplements to reach economic viability. Importantly, the increased lipid yield demonstrated here was not achieved by screening for genetic mutations. While high-dose radiation (>100 Gy) has previously been used to develop mutant microalgae strains[14], this requires difficult and laborious isolation. Furthermore, as discussed above, the low-dose radiation (1–5 Gy) treatment applied here did not induce harmful stress in the microalgae; there was no reduction in biomass or cell viability and during exponential phase the treatment increased growth rate, therefore we regard this as a beneficial, hormetic response. Toxicant-induced hormesis in algae has previously been considered, such as through low-dose application of herbicides, algicides and other chemicals[54], but use of these would increase the risk of pollution and food-chain transfer of harmful chemicals. We propose that low-dose X-rays are an ideal primer to facilitate metabolic activation; treatment time is short and the response is fast, while high-energy electromagnetic radiation allows exposure of all cells within a culture, therefore the response is very consistent and reproducible. In addition, due to different fates of ROS in the extracellular and intracellular milieu, low-dose ionizing irradiation selectively hits intracellular targets leaving no chemical trace in the aqueous environment. Moreover, unlike most chemical stress inducers, such as starvation of nitrogen or addition of chemical toxicants, a low-dose radiation primer cannot be metabolized and it does not require complicated alterations to growth media.

## Methods
### Chemicals
All chemicals were of ACS purity grade or higher. Chemicals were obtained from Sigma-Aldrich (St. Louis, MI, USA), unless otherwise stated. All experiments were performed using twice-distilled deionized water (18 MΩ) that was obtained by reagent grade water system (Millipore, Billerica, MA, USA).

### Strain, growth conditions, and irradiation protocols
*Chlorella sorokiniana* strain CCAP 211/8 K was obtained from the Culture Collection of Algae and Protozoa (CCAP), Oban, UK. *C. sorokiniana* was grown in 35 mL aliquots of 3N-BBM + V medium in 100 mL Erlenmeyer flasks that were placed on an orbital shaker (120 rpm) in a growth cabinet with temperature maintained at 22 °C, and a continuous photon flux of 120

$\mu$mol/m$^2$/s (Philips MST TL-D Reflex 36W840 1 SLV/25 lamps, Amsterdam, Netherlands). The medium was prepared according to CCAP recipe (https://www.ccap.ac.uk/wp-content/uploads/MR_3N_BBM_V.pdf), with the following composition (final concentration): 8.82 mM NaNO$_3$, 0.17 mM CaCl$_2$, 0.3 mM MgSO$_4$, 0.43 mM K$_2$HPO$_4$, 1.29 mM KH$_2$PO$_4$, 0.43 mM NaCl, trace element solution (containing 12.09 $\mu$M Na$_2$EDTA, 2.15 $\mu$M FeCl$_3$, 1.24 $\mu$M MnCl$_2$, 0.22 $\mu$M ZnCl$_2$, 0.05 $\mu$M CoCl$_2$, and 0.1 $\mu$M Na$_2$MoO$_4$), and 0.12 $\mu$g/L vitamin B$_1$ and 0.1 $\mu$g/L vitamin B$_{12}$. The initial pH of the medium was adjusted to pH 7.5. The medium was inoculated with approximately 0.5 $\times$ 10$^6$ cells/mL. The experiments were conducted in 3 steps to find the optimal time point, doses and dose rates of irradiation. In Step 1, microalgae were irradiated at 3 days after inoculation, in the early exponential phase, and the analyses were performed over 30 days. In Step 2, microalgae were irradiated at 20 days after inoculation, in the early stationary phase, and the analyses were performed at day-10 after irradiation. In Step 3, microalgae were irradiated 20 days after inoculation and detailed analyses were performed after 1 day, with some analyses also performed at day-10. In Step 2 and 3, all flasks were weighed at inoculation and the loss of water by evaporation was corrected with sterile deionized water at day-15. No more than 10% of the total volume was reduced by evaporation. Samples (35 mL) were placed in a glass Petri dish (10 cm diameter) and exposed to a single continuous dose of X-rays using a CellRad system (Faxitron Bioptics LLC, Tucson, AZ, USA), with the following settings: tube power, 750 W; filtration, 1.6 mm Be and 0.5 mm Al; energy, 120 kV. Doses and dose rates were adjusted by changing the current and were measured by a built-in ion chamber dosimeter. Samples were rotated during irradiation. Doses and dose rates of emitted radiation and radiation that was absorbed by the samples (approximately 10% of emitted doses in this setup) are presented in Supplementary Table 1. For simplicity, approximate values of absorbed doses (1, 2, 5, 10, and 20 Gy) and dose rates of irradiation (0.05, 0.25, 0.5 Gy/min) are presented in the text and the figures. Irradiated cultures were placed back into 100 mL Erlenmeyer flasks and grown under the same conditions until further analysis. Treated samples and controls (non-irradiated samples) were always grown in the same batch.

## Basic parameters of microalgal cultures

Cell growth curves were generated according to the changes in optical density that were measured at 750 nm (OD$_{750}$) with a UV/VIS spectrophotometer (Jenway Genova Plus, Stafordshire, UK), each day for 30 days following the inoculation. OD$_{750}$ values show significant linear correlation ($R^2$ = 0.975, $P$ < 0.0001) with cell counts per mL (Supplementary Fig. 10). For the measurements, samples were diluted 10$\times$ in fresh medium to keep OD$_{750}$ values below 1. Growth rates in the exponential phase of culture growth were calculated using OD$_{750}$ values at the start (day 3; X$_3$) and the end (day 20; X$_{20}$) of the exponential phase, and using the formula (ln(X$_{20}$) $-$ ln(X$_3$))/17. Cell count was carried out using a Sedgewick-Rafter counting chamber. Microalgal samples (0.5 mL) were treated with Lugol's iodine solution (10 $\mu$L) for 10 min, equally diluted to keep the number of the cells per field in the 10–50 range, and left to settle in the chamber for 5 min. Cells were counted in 10 random fields of view using an optical microscope, and cell density was calculated by multiplying mean cell number per field by dilution factor. For biomass determination, 2 mL aliquots were centrifuged at 5000 $g$ for 5 min, and the supernatant was discarded. Pellets were left to dry at 60 °C for 24 h. The pigments chlorophyll a, chlorophyll b, and carotenoids were extracted according to the following protocol. The pellet of 2 mL sample (collected as for the biomass determination) was resuspended in ice-cold methanol (2 mL), and homogenized in a glass homogenizer on ice for 2 min. The homogenate was supplemented with 8 mL of methanol and left in the dark at 4 °C for 24 h. Absorbance of the solvent extract was measured at 666 nm, 653 nm, and 470 nm. Concentrations of pigments were calculated according to Lichtenthaler and Wellburn[55], and normalized to biomass that was determined on the same day. Cell viability was established using the Evans Blue stain, by incubating cells in a 0.05% (w/v) Evans Blue solution for 15 min followed by washing in deionized water[56]. The proportion of Evans Blue stained cells corresponds to the proportion of non-viable cells. The viability is presented as a percentage of Evans Blue negative cells. At least 100 cells were analyzed per sample.

## Fluorescence assay for relative lipid content

Relative lipid content was evaluated using a rapid Nile Red assay to stain lipid droplets[57]. *C. sorokiniana* cultures were diluted 20$\times$ in 50 mM potassium phosphate buffer (pH = 7.5), so that OD$_{750}$ does not exceed 0.5. A 0.25% (v/v) bleach solution was added and samples were incubated for 1 min to minimize the interference of chlorophyll[58]. Samples were washed 2$\times$ at 2300 $\times$ $g$ for 5 min and resuspended in phosphate buffer with 25% (v/v) DMSO. Nile Red (Acros Organics, Antwerp, Belgium) was added at the final concentration of 50 $\mu$g/mL. Samples were incubated in the dark for 10 min. Fluorescence (S1/R1) was measured at 530 nm excitation and 570 nm emission using Fluorolog FL3-221 spectrofluorimeter (Jobin Yvon Horiba, Paris, France), with FluorEssence 3.5 software (Horiba Scientific, Kyoto, Japan). Fluorescence intensity was normalized to mean control fluorescence value on the same experimental day to estimate the relative lipid content in the culture (volumetric lipid content). Additionally, it was normalized to biomass to establish the relative lipid yield in biomass. Values are presented in arbitrary units. Lipid yield values determined by Nile Red fluorescence show a significant positive correlation ($R^2$ = 0.975, $P$ = 0.013) with lipid yield values determined by Soxhlet extraction and gravimetry.

## Cell volume

Cell volume was estimated from micrographs made by optical microscopy and TEM. Cross-section areas were used to establish cell radius, and the volume was calculated based on the assumption that *C. sorokiniana* cells are spherical. Aliquots (10 $\mu$L) were smeared on microscopic slides and 5 micrographs (40$\times$ magnification) per slide were randomly collected. Micrographs were analyzed in ImageJ image processing software (National Institutes of Health, USA). Cross-section areas were determined using the following ImageJ macro that was optimized by comparison with hand-select analysis: Import image; Image-Type-8-bit; Process-FFT-Bandpass Filter (100 px; 3 px; None; 5%; Autoscale, Saturate); Process-Find Edges; Process-Binary-Make Binary; Process-Binary-Fill Holes; Analyze Particles (200–Infinity; Pixel; Circularity 0.8–1.0). In TEM micrographs (7500$\times$ magnification), cross-section areas were established in ImageJ by hand-selected analysis of >25 cells with the nuclear mid-section for each sample.

## Starch content

Relative starch content in microalgal cultures was estimated using a rapid assay based on Lugol's staining of starch[59]. Microalgal culture samples (200 $\mu$L) were placed into 96-well microplates. Lugol's solution (5 $\mu$L) was added and the suspensions were mixed. Each stained sample had an unstained pair. Optical density at 660 nm (OD$_{660}$) was measured using a microplate reader. Relative starch content was calculated as follows: OD$_{\text{Lugol stained sample}}$ - OD$_{\text{unstained sample}}$ - OD$_{\text{Lugol}}$ - OD$_{\text{water}}$. The obtained values were further normalized to the mean control value, and to biomass to measure relative starch yield. The results were presented in arbitrary units. Next, to determine absolute starch yield, microalgal culture samples (4 mL) were centrifuged at 5000 $g$ for 5 min, pellets were dried at 60 °C for 24 h, and resuspended in 80% (v/v) ethanol (500 $\mu$L). Biomass was homogenized by 4 $\times$ 15 s mixing at 30 Hz with 5 mm stainless steel beads in mixer mill (MM400, Retsch, Haan, Germany). The samples were then incubated in 80% (v/v) ethanol at 85 °C for 5 min to remove the pigments (the procedure was repeated until the pellet became colorless). Starch yield in biomass was determined with a Total Starch Assay Kit (Megazyme International Ireland Ltd., Wicklow, Ireland), according to the modified AOAC 996.11 method[60]. Pellets were resuspended in a mixture of 80% (v/v) ethanol (200 $\mu$L) and DMSO (500 $\mu$L), and incubated at 90 °C for 1 h. Thermostable $\alpha$-amylase was diluted 30-fold in 50 mM MOPS buffer (pH = 7) with 5 mM CaCl$_2$ and added to samples (30 units). The samples were incubated at 90 °C for 15 min. Finally, 500 $\mu$L of 50 mM sodium acetate buffer (pH = 4,5) with 5 mM CaCl$_2$ and 10 $\mu$L of amyloglucosidase (33 units) was added and the samples were incubated at 50 °C for 1 h. To measure released glucose,

supernatant was separated from the pellet by centrifugation $13000 \times g$ for 10 min, mixed with GOPOD reagent from the kit in 1:5 ratio, and incubated at 50 °C for 20 min. Samples were cooled to room temperature and absorbance was measured at 508 nm. Supernatant was diluted prior to mixing with GOPOD reagent to fit the standard curve that was made using a serial dilution of glucose solution from the kit. Concentrations were calculated from the standard curve.

## Lipid yield and fatty acids profile analysis

Samples were centrifuged at $5000 \times g$ for 5 min, and the pellets were left to dry at 50 °C for 24 h. Biomass was homogenized with mortar and pestle, and samples were merged to obtain at least 170 mg of biomass for each extraction. Lipids were extracted using Soxhlet (SOX406 Semi-Automatic Soxhlet Fat Analyzer, Hanon, Beijing, China), with the following settings: solvent mixture, chloroform:methanol 2:1 (v/v); temperature, 80 °C; time, 4 h. Before and after the extraction, samples were dried at 50 °C for 12 h and cooled in desiccator with silica gel. The analysis of fatty acid methyl ester (FAME) profiles involved transesterification through acidic methanolysis and gas chromatography-mass spectrometry (GC-MS). Extracted lipids (30 mg) were dissolved in 6 mL of methanol with 2-3 drops of concentrated sulfuric acid. The mixture was refluxed at 80 °C for 2 h and then pH was adjusted to 7 using $NaHCO_3$ solution (0.1 g/mL of water). FAMEs were collected using hexane ($4 \times 6$ mL). The hexane layer was collected with a Pasteur pipette, and dried with 15 g of anhydrous $Na_2SO_4$ for 15 min. The solution was filtered to remove the drying agent, and the solvent is removed in a rotary film evaporator working at 40 °C under reduced pressure. The FAME extract was dissolved in dichloromethane (5 mg/mL) and purified by vigorous mixing with activated charcoal (20 mg/mL) and Sephadex A25 (6 mg/mL). The analysis was performed using a GC-MS QP2010 plus, equipped with an AOC 5000 injector (Shimadzu, Kyoto, Japan), and FAME column (Phenomenex, L = 30 m, ID = 0.25 mm, df = 0.50 μm), and using GCMSsolution Ver. 2 software (Shimadzu). Samples (1 μL) were injected in the split mode (1:30), with the injector temperature set to 250 °C. Mass spectra were acquired in EI mode ( ± 70 eV) in the m/z range 50–500 amu (SCAN) mode. Helium (99.999%) was used as a carrier gas with a flow rate of 1.34 mL/min. The column was heated linearly from 100 °C (hold 2 min) to 240 °C with a gradient of 3 °C/min and hold at 240 °C for 5 min. Ion source temperature was set to 240 °C; interface temperature to 260 °C. Identification of constituents was performed by comparing their mass spectra to those from NIST05, Wiley8 and FFNSC3 libraries, using different search engines, and a set of FAME standards in Supelco® 37 Component FAME Mix that was dissolved in hexane (1 mg/ mL). Quantitative data were obtained from GC peak area by the method of area normalization then the results were expressed relative to cell biomass. Cetane numbers were calculated using the formula and reference numbers presented by Knothe[61]. Cetane numbers for C15:1, C17:0, and C16:3 are not available and were taken to be the same as for C16:1, C16:0, and C18:3, respectively.

## TEM analysis

*C. sorokiniana* cells were collected by centrifugation at $5000 \times g$ for 5 min and fixed overnight in 0.1 M phosphate buffer (pH = 7.2) containing 3% (v/v) glutaraldehyde (Serva, Heidelberg, Germany) and 1% (v/v) paraformaldehyde (pH = 6.9) at 4 °C. Post-fixation was performed with 1% (w/v) osmium tetroxide (Serva) in 0.1 M phosphate buffer (pH = 7.2) at room temperature for 2 h. Samples were dehydrated in a graded acetone series and embedded in resin for soft blocks (AGR1031, Agar Scientific, Stansted, UK). Thin sections (70 nm), obtained with a Leica UC7 ultramicrotome (Leica Microsystems, Wetzlar, Germany), were stained with uranyl acetate and lead acetate and observed at 60 kV using a JEOL JEM-1010 TEM (JEOL, Tokyo, Japan) with an XR16 CCD camera and AMT Image Capture Engine (Advanced Microscopy Techniques, Woburn, MA, USA). The analysis of micrographs that included cell cross-section area, number of lipid droplets and starch granules per cell cross-section, and the total areas of all lipid droplets and starch granules in the cross-sections, was performed using ImageJ. At least 25

randomly selected cells with the nuclear mid-section in 3 independent replicates were analyzed for each treatment.

## EPR spectroscopy

*C. sorokiniana* cells were collected by centrifugation at $5000 \times g$ for 5 min and washed 3× with water to remove extracellular EPR-active metals. Each cell pellet (100 mg) was mixed with 100 μL of water, placed into quartz cuvettes (Wilmad-Lab Glass, Vineland, NJ, USA), and quickly frozen in cold isopentane. EPR spectra were recorded at 19 K on a Bruker Elexsys II E540 spectrometer with XEPR software, operating at X-band (9.4 GHz), with an Oxford Instruments ESR900 helium cryostat, at the EPR Laboratory, Faculty of Physical Chemistry, University of Belgrade. The experimental parameters were: microwave power, 3.17 mW; scan time, 2 min; modulation amplitude, 0.5 mT; modulation frequency, 100 kHz; number of accumulations, 4. Signal amplitude *vs.* power plot was built to establish power range that avoids saturation. All spectra were baseline corrected.

## Transcriptomic analysis

*C. sorokiniana* cells were collected by centrifugation at $5000 \times g$ for 5 min, washed 3× with ice-cold deionised water, snap-frozen in liquid $N_2$, and stored at −80 °C until further analysis. RNA extraction was performed by the addition of 1 mL TRIzol Reagent (Thermo Fisher Scientific, Waltham, MA USA), which was added to each sample and left to incubate at room temperature for 5 min. Chloroform (200 μL) was added to each sample for further extraction. Samples were briefly vortexed and left to stand for additional 5 min. To isolate RNA from DNA, proteins and other cell debris, the samples were centrifuged for 15 min at $12000 \, g$ and 4 °C. The upper layer was then aliquoted, taking care not to disturb the lower layers. To precipitate the RNA, 500 μL of isopropanol was added to each sample and left for 15 min at room temperature. To collect the precipitated RNA, samples were centrifuged for 10 min at $12000 \, g$ and 4 °C and the supernatant was removed. The RNA pellet was washed two times with 1 mL of 75% ethanol (v/v) and centrifugation at $12,000 \times g$ for 5 min at 4 °C. Samples were left on ice for 30 min to allow any remaining ethanol to evaporate. The RNA pellet was then resuspended in 50 μL sterile deionized $H_2O$. The total RNA samples were submitted to the Faculty of Biology, Medicine and Health Genomic Technologies Core Facility, University of Manchester, for cDNA library preparation and sequencing. Quality and integrity of the RNA samples were assessed using a 2200 TapeStation (Agilent Technologies, Santa Clara, CA, USA) and then libraries were generated using the TruSeq® Stranded mRNA assay (Illumina, Inc., San Diego, CA, USA) according to the manufacturer's protocol. RNA samples with a RIN value > 7 were used for library generation. RIN values were: Control-1, 8.7; Control-2, 9.4; Control-3, 9.1; Irradiated-1, 8.2; Irradiated-2, 7.1; Irradiated-3, 7.8. Briefly, total RNA (0.1–4 μg) was used as input material from which polyadenylated mRNA was purified using poly-T, oligo-attached, magnetic beads. The mRNA was then fragmented using divalent cations under elevated temperature and then reverse transcribed into first strand cDNA using random primers. Second strand cDNA was then synthesized using DNA Polymerase I and RNase H. Following a single 'A' base addition, adapters were ligated to the cDNA fragments, and the products were then purified and enriched by PCR to create the final cDNA library. Adapter indices were used to multiplex libraries, which were pooled prior to cluster generation using a cBot instrument. The loaded flow-cell was then paired-end sequenced (76 + 76 cycles, plus indices) on an Illumina HiSeq4000 instrument. Finally, the output data was demultiplexed (allowing one mismatch) and BCL-to-Fastq conversion performed using Illumina's bcl2fastq software, version 2.20.0.422. All sequence reads were pre-processed using the Trimmomatic filter software[62] to remove adapters and contaminants from the data. After the data was cleaned, reads were mapped and counted to a reference genome assembly (version 2) of *C. sorokiniana* UTEX 1602 (https://www.ncbi.nlm. nih.gov/assembly/GCA_002245835.2) generated by Arriola et al. [23]. Mapping was performed using STAR mapping software[63] whilst the read counting was performed using the htseq-count script tool in HTSeq

software[64]. Finally, normalization and differential expression calculations were performed using DESeq2 software[65]. Transcript abundance was presented as normalized counts derived from the DEseq2. Heatmaps were generated using Morpheus software (software.broadinstitute.org/Morpheus) and clustered using k-means analysis, which allowed demonstration of strong clustering between independent replicate samples, showing that the transcriptional changes and the metabolic response to low-dose irradiation was reproducible (Supplementary Fig. 11). The *C. sorokiniana* gene transcript annotation data was obtained from JGI PhyCosm (https://phycocosm.jgi.doe.gov/Chloso1602_1/Chloso1602_1.home.html). For all transcripts that showed significant differential expression between the irradiated versus control treatments (false discovery rate (FDR) < 0.05; with a FDR adjusted *p* value generated as described[66]), transcript annotation was further manually validated, including by use of BLASTx comparisons with annotated sequences from the *Chlamydomonas reinhardtii* CC-4532 v6.1 genome annotation (https://phytozome-next.jgi.doe.gov/info/CreinhardtiiCC_4532_v6_1), using the BLAST tools on the JGI Phytozome genomics portal. KEGG annotation (https://www.genome.jp/kegg/annotation/) was used to further determine functional classes (Supplementary Data 1), while KEGG Mapper (https://www.genome.jp/kegg/mapper/)[67] was used to map transcripts to lipid metabolism pathways (Supplementary Fig. 7 and Supplementary Data 1). Multiple sequence alignments to determine distinct gene isoforms were performed using translated amino acid sequences and Clustal Omega (https://www.ebi.ac.uk/Tools/msa/clustalo/) using default settings. Conserved amino acids of CrPSR1 and the *C. sorokiniana* PSR1 orthologue (C2E21_4446) were visualized using Easy Sequencing in PostScript (ESPript; https://espript.ibcp.fr/ESPript/ESPript/)[68] with standard default parameters.

## Quantitative reverse transcription PCR (qPCR)

RNA was extracted from *C. sorokiniana* cells as described above. RNA was treated with RQ1 DNase (Promega, Madison, WI, USA) and cDNA synthesis was performed using a Superscript III reverse transcriptase kit (Thermo Fisher Scientific, Waltham, MA USA) and an oligo(dT) primer (Promega, Madison, WI, USA). The qPCR was prepared using 100 ng of cDNA in triplicate (technical replicates) of three biological replicates of the control and irradiated samples in a 20 μL sample containing 10 μL of SensiFAST SYBR Hi ROX kit (Meridian Bioscience, Cincinnati, OH, USA) and 1 mM of each oligonucleotide primer (Eurofins Genomics, Ebersberg, Germany). Primer sequences are shown in Supplementary Table 2. The reaction was performed using a StepOnePlus™ Real-Time PCR machine with StepOne™ software v2.3. The *C. sorokiniana* 18 S rRNA gene (GenBank accession number KR904895) was used as a normalization control gene. Relative gene expression was determined using the $2^{-\Delta\Delta CT}$ method[69].

## Statistics and reproducibility

All experiments were performed in at least biological triplicates. The exact numbers of biological replicates in different experiments are described in each figure and are available alongside the raw source data shown in the Source Data file (Supplementary Data 2). Values are presented as means ± standard error. Differences between treated samples and controls were tested using a non-parametric two-tailed or one-tailed Mann–Whitney U test, as appropriate. Results were considered to be statistically significant if $p < 0.05$. All individual $p$ values are listed in Supplementary Data 2. Statistical analysis was performed in STATISTICA 8.0 (StatSoft Inc., Tulsa, OK, USA). $OD_{750}$ data were fitted using sigmoidal fit. The goodness of fits was evaluated by $R^2$ (the adjusted $R$-square value), which was > 0.95 for all sets of analyzed data.

## Ethics and inclusion statement

The author list includes contributors from the locations where the research was conducted, who participated in study conception, study design, data collection, analysis, and interpretation of the findings.

## Reporting summary

Further information on research design is available in the Nature Portfolio Reporting Summary linked to this article.

## Data availability

Sequence data from this article can be found in the EMBL-EBI ArrayExpress data library under accession number E-MTAB-12288. All raw source data are available in Supplementary Data 1 and in Supplementary Data 2 (Source Data file).

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

## Acknowledgements

This work was supported by The NATO Science for Peace and Security Program (project grant number G5320 to B.Z., A.M.L., I.S. and J.K.P.). This work was also supported in part by a UK Biotechnology and Biological Sciences Research Council (BBSRC) PhD studentship (grant number BB/M011208/1 to O.A.M.), and by the Ministry of Science, Technological Development and Innovation of the Republic of Serbia (Grant No. 451-03-9/2021-14/200053). We thank Andy Hayes and Claire Morrisroe in the University of Manchester Genomic Technologies Core Facility for performing RNA-Seq sequencing and processing, and Leo Zeef for performing sequence read mapping and bioinformatics support.

## Author contributions

I.S., J.K.P, B.Z. and A.M.L. conceived and designed the study. M.S., M.J., S.K., M.D., J.D.L. and I.S. performed microalgae physiology and biochemistry experimentation and data analysis. B.Z. and J.D.L. performed TEM analysis. J.K.P. and O.A.M. performed gene transcriptomics analysis. J.K.P. and I.S. prepared the manuscript, with support from all authors.

## Competing interests

The authors declare no competing interests.
