## [Peer Review File · Communications Biology]

Reviewers' comments:

Reviewer #1 (Remarks to the Author):

The purpose of the present work was to analyze the effect of low-doses (1–5 Gy) of ionizing radiation in the X-ray range, on *Chlorella sorokiniana* (a green microalga) in terms of biological and biochemical responses. Authors have established that low-doses of ionizing radiation can induce a non-toxic and rapid (24hs) hormetic response, promoting the up-regulation of > 30 lipid metabolism genes, such as genes encoding an acetyl-CoA carboxylase subunit, phosphatidic acid phosphatase, lysophosphatidic acid acyltransferase, and diacylglycerol acyltransferase.

The paper is well written and designed. In addition, the experimental work is original and results interesting. Thus, it should be considered for publication, however, some weaknesses, listed below, should necessarily be addressed.

Materials and methods:

Strain, growth conditions, and irradiation protocols section; Line 321: “all flasks were weighed at inoculation and the loss of water by evaporation was corrected with sterile deionized water at day-15.” Why did authors use water instead of medium? Which percentage of the total volume represented this evaporation? This is an important issue as the water added could alter the whole experiment in terms of nutrient availability.

Results:

According to Borowitzka (2018), and with reference to microalgae, “stress is defined as the disruption of homeostasis due to a stressor. Then, the stress response represents the changes in cell metabolism during acclimation and restoration of homeostasis.” According to this definition, authors must consider ionizing radiation as a stressor, that it is applied at the early stationary phase to avoid its negative effect on microalgal growth. Also, the induction of the genes involved in lipid metabolism (including the changes reported in lipid droplets) are those typically observed as the cellular response to stress.

Discussion:

Line 270: “The work described, provides a template, validated here for *C. sorokiniana*, for a non-invasive method to induce a rapid and reproducible increase in lipid yield at the point of microalgal biomass harvesting. Such innovation is required to allow mass cultivation of microalgae for applications including biofuel and nutritional supplements to reach economic viability.” How do authors think that ionizing radiation could be applied to massive microalgal cultivation? In this way, authors clearly stated in the Materials and Methods section that experiments were performed in 35 mL final volume.

Reviewer #2 (Remarks to the Author):

In this manuscript, Dr. Pittman et al. describe the use of low dosages of radiation to improve lipid content of *Chlorella sorokiniana* algal cells without impacting biomass production. Three subsequent experiments were conducted to explore how exposure to different levels of radiation across different time points (exponential, and early stationary) impacts the physiological characteristics, growth rates, and transcriptomic profile of algal cultures. Lipid content was

measured using both Nile Red fluorescence as a proxy, and GCMS for absolute lipid content. Starch content was measured using a colorimetric assay. mRNA was extracted from cultures and sequenced using an Illumina HiSeq and analyzed data was used to make conclusions regarding the impact of radiation on the metabolic pathways of *Chlorella sorokiniana*. Physiological and transcriptomic data were complemented by TEM and spectroscopy.

Overall, the manuscript is well-written and easy to follow. Further, I believe it makes meaningful contributions to the field of algal bioproduct production with implications for global sustainability. To my knowledge, this is the first report of an external treatment that induced lipid accumulation in algae without negatively impacting growth and biomass production. Many approaches to improving biomass production in algae focus on inducing different stressors and this manuscript will be useful and applicable to most algal cultivation systems. From this perspective, the manuscript will be of interest to the readership of *Communications Biology*. I find that the overall conclusion of this manuscript is valid and supported by the data provided. However, a few issues need to be addressed prior to publication.

Below are my comments

110 The authors claim that Low-dose irradiation during the exponential phase increases cell growth rate and biomass. However, there are only two time points that represent a significant increase in biomass production of cultures exposed to radiation (Day 30 with 2 Gy at 0.05 Gy/min, and Day 25 with 1 Gy at 0.25 Gy/min).

I am concerned that the language/text of this section is over-interpreting the results of step 1 cultures and suggesting an overall trend of increased biomass after low radiation exposure while this is not actually represented in the data.

115 The authors see an increase in volumetric lipid content in the 1 Gy at 0.25 114 Gy/min and 5 Gy at 0.5 Gy/min treatments at day 30 and claim that “This higher total lipid content was due to the increased biomass production.” However, neither of these conditions show a significant difference in biomass production to the control on day 30.

Nile Red fluorescence is a decent proxy for lipid production; however, absolute values using the GCMS is much more convincing. If biomass is available from step 1 and 2 cultures, I would suggest performing a transesterification and GCMS analysis on these conditions to improve the confidence of this analysis. I understand that these steps were likely screening steps, but this analysis would strengthen the conclusions.

Do you have error bars for lipid speciation in figure 2 D? Statistical analysis would strengthen the data presented on lines 147-169.

162 These data suggest that lipids content is increased while starch content is decreased in cultures that are exposed to radiation. The authors claim that this supports a “mobilization of starch to lipids” that is supported by previous literature. This claim could be further supported by using the transcriptomic analysis presented in this paper. Expanding the transcriptomic analysis to

explore the differential expression of starch degradation and associated pathways would significantly strengthen this claim.

164 “there is some increase in carbohydrate content in *C. sorokiniana* cells following irradiation, in the form of cell wall polymers that have previously been observed to give rise to a thicker cell wall and increased cell wall yield in response to 1–5 Gy of X-ray treatment” what data/figure is this claim associated with?

198 What is your justification for using a log₂ fold change cutoff of 0.5? In my experience cutoff values range from 1-2.

Reviewing the expression of stress mitigation pathways and seeing if there are transcriptional stress responses to the low dose of radiation would be another method to confirm that radiation is not negatively impacting the algal culture in addition to the growth analysis presented in this paper. This review paper (<https://doi.org/10.1016/j.toxrep.2019.10.001>) discusses common pathways that algae use to mitigate ROS that might be of interest to look at.

The materials and methods section is well-written and reproducible. It was very clear how all analyses were done within this manuscript and this aided in the flow of the manuscript and reviewing process. Below are a few comments/suggestions regarding the methods.

322 I would specify what size petri dish was used for this analysis.

422 Was an internal standard used during the lipid extraction and transesterification process and if not now did the authors account for potential errors introduced during the extraction, purification, and analysis of lipids?

Figure 1 second line: of irradiation ◊ of irradiation

Fig 1 a. Are these growth curves based on real data? I would clarify if they are not.

Reviewer #3 (Remarks to the Author):

The authors investigated the effects of low-doses of ionizing radiation in the X-ray range on microalga *Chlorella sorokiniana* by analyzing the lipid, Starch, fatty acids contents, followed by transcriptomic analysis to reveal the molecular mechanism. Overall, the study is well-designed and the results are valuable for future biotechnology application. However, the main concern is the selection of differentially expressed genes (DEGs) in bioinformatic analysis, this must be solved prior to further consideration for publication.

Specific comments:

1. Line 530: a $p < 0.05$ was used as a cutoff to determine the DEGs for bioinformatic analysis.

However, in transcriptomic analysis this can give large numbers of false positives. Hence, q value (adjusted p value) is usually calculated and applied for determining DEGs in RNA sequencing. Search 'P and q values in RNA Seq' and find the detailed difference. Anyway, the DEGs need to be determined by q value rather than p value.

2 Line 500 'Quality and integrity of the RNA samples were assessed using a 2200 TapeStation'

Please specify the quality assessment results of RNA samples, e.g., RIN number.

3 Line 548: 'Differences between treated samples and controls were tested using a non-parametric two-tailed Mann-Whitney U test'. In statistical analysis, a normality test is usually performed, where data passed it an one-way ANOVA is applied, otherwise non-parametric analysis is used.

Please justify why non-parametric test was applied for all results without normality test.

We thank the editor and reviewers for their constructive comments in improving our manuscript. Our point-by-point responses to each comment are provided below.

Reviewer #1:

Comment 1	Response 1
The purpose of the present work was to analyze the effect of low-doses (1–5 Gy) of ionizing radiation in the X-ray range, on Chlorella sorokiniana (a green microalga) in terms of biological and biochemical responses. Authors have established that low-doses of ionizing radiation can induce a non-toxic and rapid (24hs) hormetic response, promoting the up-regulation of > 30 lipid metabolism genes, such as genes encoding an acetyl-CoA carboxylase subunit, phosphatidic acid phosphatase, lysophosphatidic acid acyltransferase, and diacylglycerol acyltransferase. The paper is well written and designed. In addition, the experimental work is original and results interesting. Thus, it should be considered for publication, however, some weaknesses, listed below, should necessarily be addressed.	We thank the reviewer for their positive opinion of our work.
Comment 2	Response 2
Materials and methods: Strain, growth conditions, and irradiation protocols section; Line 321: “all flasks were weighed at inoculation and the loss of water by evaporation was corrected with sterile deionized water at day-15.” Why did authors use water instead of medium? Which percentage of the total volume represented this evaporation? This is an important issue as the water added could alter the whole experiment in terms of nutrient availability.	We understand the Reviewer's criticism. We added water because it is mainly water that evaporates (not the whole medium), which makes the micronutrient medium slightly more concentrated. However, the main reason for this correction is that evaporation varies and may affect measurements of biomass yield. The evaporation represents no more than 10% of the total volume, and as such, we added 1.5 to 3 mL per 35 mL sample. This maximum volume reduction and adjustment is now clarified in the revised methods section (lines 379-380).
Comment 3	Response 3
Results: According to Borowitzka (2018), and with reference to microalgae, “stress is defined as the disruption of homeostasis due to a stressor. Then, the stress response represents the changes in cell metabolism during acclimation and restoration of homeostasis.” According to this definition, authors must consider ionizing radiation as a stressor, that it is applied at the early stationary phase to avoid its negative effect on microalgal growth. Also, the induction of the genes involved in lipid metabolism (including the changes reported in lipid droplets) are those typically observed as the cellular response to stress.	We agree with the reviewer that ionizing radiation is indeed a stressor, and that the lipid responses observed are consistent with responses to nutrient starvation stresses. Indeed, we have stated in the manuscript that “Ionizing radiation can induce stress through radiolytic production of ROS that may result in a metabolic response” (lines 186-187). Furthermore, we have now added a new paragraph in the Discussion section (from line 313) to clarify the point that the responses observed are consistent with previously studied stress responses in terms of metabolic changes. However, we clarify that a key difference with the ionizing radiation stressor at these low doses is that this does not induce a significant inhibition in cell growth.

Comment 4	Response 4
Discussion: Line 270: "The work described, provides a template, validated here for C. sorokiniana, for a non-invasive method to induce a rapid and reproducible increase in lipid yield at the point of microalgal biomass harvesting. Such innovation is required to allow mass cultivation of microalgae for applications including biofuel and nutritional supplements to reach economic viability." How do authors think that ionizing radiation could be applied to massive microalgal cultivation? In this way, authors clearly stated in the Materials and Methods section that experiments were performed in 35 mL final volume.	We appreciate the Reviewer's question. We are now working on the development and scale-up of the technology, and performing validation of the responses at larger volumes, although these results will be out of the scope for this current study. Large X-irradiation units for field use are already available on the market (e.g. for pipe welding checks), so we believe that a short one-time per cycle irradiation of larger volumes of microalgal culture passing through tubing surrounded by a sufficiently shielded X-ray source at controlled volumes, speed and retention times to achieve the required dose, should be an achievable process.

Reviewer #2:

Comment 1	Response 1
In this manuscript, Dr. Pittman et al. describe the use of low dosages of radiation to improve lipid content of Chlorella sorokiniana algal cells without impacting biomass production. Three subsequent experiments were conducted to explore how exposure to different levels of radiation across different time points (exponential, and early stationary) impacts the physiological characteristics, growth rates, and transcriptomic profile of algal cultures. Lipid content was measured using both Nile Red fluorescence as a proxy, and GCMS for absolute lipid content. Starch content was measured using a colorimetric assay. mRNA was extracted from cultures and sequenced using an Illumina Hiseq and analyzed data was used to make conclusions regarding the impact of radiation on the metabolic pathways of Chlorella sorokiniana. Physiological and transcriptomic data were complemented by TEM and spectroscopy. Overall, the manuscript is well-written and easy to follow. Further, I believe it makes meaningful contributions to the field of algal bioproduct production with implications for global sustainability. To my knowledge, this is the first report of an external treatment that induced lipid accumulation in algae without negatively impacting growth and biomass production. Many approaches to improving biomass production in algae focus on inducing different stressors and this manuscript will be useful and applicable to most algal cultivation systems. From this perspective, the manuscript will be of interest to the readership of Communications Biology. I find that the overall conclusion of this manuscript is valid and supported by the data provided. However, a few issues need to be addressed prior to publication.	We thank the reviewer for their positive opinion of our work.

Comment 2 110 The authors claim that Low-dose irradiation during the exponential phase increases cell growth rate and biomass. However, there are only two time points that represent a significant increase in biomass production of cultures exposed to radiation (Day 30 with 2 Gy at 0.05 Gy/min, and Day 25 with 1 Gy at 0.25 Gy/min). I am concerned that the language/text of this section is over-interpreting the results of step 1 cultures and suggesting an overall trend of increased biomass after low radiation exposure while this is not actually represented in the data.	Response 2 We understand this concern and have now revised the text of this section to highlight that indeed only two time points represented a significant increase in biomass production (lines 115-116). We agree that while the data does not suggest an overall trend of increased biomass after low radiation exposure, it does suggest a trend of increased growth rate after low radiation exposure during the exponential phase.
Comment 3 115 The authors see an increase in volumetric lipid content in the 1 Gy at 0.25 Gy/min and 5 Gy at 0.5 Gy/min treatments at day 30 and claim that “This higher total lipid content was due to the increased biomass production.” However, neither of these conditions show a significant difference in biomass production to the control on day 30.	Response 3 We have revised the text to simply state that there was an increase in volumetric lipid content in these treatment conditions and no longer make this claim about higher lipid content being due to increased biomass (lines 120-121).
Comment 4 Nile Red fluorescence is a decent proxy for lipid production; however, absolute values using the GCMS is much more convincing. If biomass is available from step 1 and 2 cultures, I would suggest performing a transesterification and GCMS analysis on these conditions to improve the confidence of this analysis. I understand that these steps were likely screening steps, but this analysis would strengthen the conclusions.	Response 4 Unfortunately we no longer have sufficient biomass available from these initial cultures to perform absolute lipid quantification by gravimetry (Soxhlet method) or by GC-MS. However, for some samples (shown in Fig. 2a and new Supplementary Fig. 1c – see below) we did perform both Nile red fluorescence quantification and Soxhlet extraction and find a positive, significant linear correlation between lipid quantification by both methods ($R^2=0.975$, $P=0.013$), indicating a positive relationship between the methods and providing confidence in the Nile red approach for initial screening (described in the Methods in lines 438-440). Furthermore, we would argue that since these initial measurements were simply for screening of X-ray treatment conditions and that we have subsequently validated the lipid production in response to the selected X-ray treatment by GC-MS (step 3), all of the conclusions concerning lipid production remain unchanged.

New Supplementary Fig. 1c:

Supplementary Figure 1. (c) Relative lipid content in the culture at 24 h after irradiation that was applied in the early stationary phase of culture growth. Relative lipid content was determined by Nile Red fluorescence (a.u., arbitrary units). All data are means \pm standard error ($n = 3-16$). Significant difference compared to control is indicated at $P < 0.05$ (*).

Comment 5 Do you have error bars for lipid speciation in figure 2 D? Statistical analysis would strengthen the data presented on lines 147-169.	Response 5 This fatty acid analysis has been repeated and the data replotted as mg of fatty acids per g of biomass with error bars provided in a revised Fig. 2d – see below. Statistical analysis has now been performed on this data and we found significant increases in specific fatty acids in response to the 1 Gy at 0.25 Gy/min treatment. This detail is now described in the revised results text to strengthen the data presented (lines 156-158 and 164-166).
Revised Fig. 2d:  Figure 2. (d) Fatty acids profile with fractions of saturated (SFA), mono-unsaturated (MUFA), and poly-unsaturated fatty acids (PUFA), and cetane values of total fatty acids (shown inset). All data are means \pm standard error (n = 4-8). Significant difference compared to control (non-irradiated) treatment is indicated at P < 0.05 (*).	
Comment 6 162 These data suggest that lipids content is increased while starch content is decreased in cultures that are exposed to radiation. The authors claim that this supports a “mobilization of starch to lipids” that is supported by previous literature. This claim could be further supported by using the transcriptomic analysis presented in this paper. Expanding the transcriptomic analysis to explore the differential expression of starch degradation and associated pathways would significantly strengthen this claim.	Response 6 We did originally examine the transcriptomic dataset to determine whether there was any evidence to support a reduction in levels of transcripts associated with starch metabolism but none of the key enzymes in starch biosynthesis such as the starch synthases or in starch catabolism such as alpha amylase or starch debranching enzyme showed any significant change in gene expression. However, clearly there is no significant increase in starch biosynthesis genes at the same time that many of the genes encoding lipid biosynthesis do increase. We have added a statement in the discussion to further clarify this point (lines 320-323).
Comment 7 164 “there is some increase in carbohydrate content in C. sorokiniana cells following irradiation, in the form of cell wall polymers that have previously been observed to give rise to a thicker cell wall and increased cell wall yield in response to 1–5 Gy of X-ray treatment” what data/figure is this claim associated with?	Response 7 This data is from one of our previous studies (Vojvodić, et al., 2020 Chemosphere) that we do cite at the end of this sentence. We have now made it clearer that we are referring to data in this referenced study (lines 175-177).

Comment 8

198 What is your justification for using a log2 fold change cutoff of 0.5? In my experience cutoff values range from 1-2.

Response 8

In our view it is also informative to consider transcripts that displayed a lower range of expression change, although more important is whether the differential expression is statistically significant. Therefore, in addition to significant DEGs with a Log2 FC >1 or <-1, we were also keen to understand those with a lower degree of expression change, so between 0.5 and 1 (or -0.5 and -1). The idea was not to handpick specific genes with small increments in expression but to see if there was a systematic response in some metabolic pathways. This can be seen with lipid metabolism where a high number of gene expression changes were observed, some with large expression increases and others with more moderate yet significant increases. Of the 410 DEGs that were significant with a FDR <0.05 and Log2 FC >0.5 or <-0.5, 193 had a Log2 FC between 0.5 and 1 (or -0.5 and -1). We have clarified this point in the text (lines 212-215).

In addition, we have performed qPCR on selected transcripts for further validation, including eight of the lipid genes (Fig. 3c) presented in the schematic in Fig. 3d, and three transcription factor genes (Fig. 3e) and four ROS response genes (Fig. 3f) – see below. In all cases, the increase in transcript abundance relative to the control treatment was significantly different and had a Log2 FC >1.

New Fig. 3c, e and f:

Figure 3 (c) Relative expression of selected lipid metabolism genes in response to irradiation as determined by qPCR and shown relative to the control (non-irradiated) expression. Numbers in square brackets represent isoforms. (e) Relative expression of three transcription factor genes as determined by qPCR and shown relative to the control (non-irradiated) expression. (f) Relative expression of four reactive oxygen species response genes as determined by qPCR and shown relative to the control (non-irradiated) expression. All data (in c, e & f) are means \pm standard error ($n = 3$) and the differences compared to control are all significant at $P < 0.05$.

Comment 9	Response 9
Reviewing the expression of stress mitigation pathways and seeing if there are transcriptional stress responses to the low dose of radiation would be another method to confirm that radiation is not negatively impacting the algal culture in addition to the growth analysis presented in this paper. This review paper (https://doi.org/10.1016/j.toxrep.2019.10.001 [doi.org]) discusses common pathways that algae use to mitigate ROS that might be of interest to look at.	We identified four transcripts that encode putative ROS response enzymes that are involved in ROS mitigation or redox response, a glutaredoxin, a thioredoxin, a catalase and an ascorbate peroxidase, and confirmed significant increase in expression following irradiation by using qPCR (shown in Fig. 3f – see above). This further supports the conclusion that generation of ROS by X-ray mediated radiolysis is both sensed by the cell and in turn elicits redox signaling, and is scavenged via antioxidant enzymes, and as such this maintains cell viability. These new results and this point has now been further discussed in the manuscript (lines 278-290).
Comment 10	Response 10
The materials and methods section is well-written and reproducible. It was very clear how all analyses were done within this manuscript and this aided in the flow of the manuscript and reviewing process. Below are a few comments/suggestions regarding the methods. 322 I would specify what size petri dish was used for this analysis.	The details of the petri dishes used (glass petri dish with 10 cm diameter) has been added to the methods section (lines 380-381).
Comment 11	Response 11
422 Was an internal standard used during the lipid extraction and transesterification process and if not how did the authors account for potential errors introduced during the extraction, purification, and analysis of lipids?	We understand the Reviewer's criticism. We optimized the method of extraction first, testing different temperatures and times of extraction. The applied method delivered the highest yield from test samples. We have not used an internal standard. There may be some losses in the process. However, the emphasis was on the comparison and not on absolute values.
Comment 12	Response 12
Figure 1 second line: of irradiation □ of irradiation	Thank you for spotting - this typo has been corrected.
Comment 13	Response 13
Fig 1 a. Are these growth curves based on real data? I would clarify if they are not.	We have now clarified in the Fig. 1 legend that the Fig. 1a growth curves are illustrative as part of a schematic and not real data (line 838).

Reviewer #3:

Comment 1	Response 1
The authors investigated the effects of low-doses of ionizing radiation in the X-ray range on microalga Chlorella sorokinianato by analyzing the lipid, Starch, fatty acids contents, followed by transcriptomic analysis to reveal the molecular mechanism. Overall, the study is well-designed and the results are valuable for future biotechnology application. However, the main concern is the selection of differentially expressed genes (DEGs) in bioinformatic	We thank the reviewer for their positive view of the work and are pleased to address the concern regarding DEG selection.

analysis, this must be solved prior to further consideration for publication.

Comment 2

1. Line 530: a $p < 0.05$ was used as a cutoff to determine the DEGs for bioinformatic analysis. However, in transcriptomic analysis this can give large numbers of false positives. Hence, q value (adjusted p value) is usually calculated and applied for determining DEGs in RNA sequencing. Search 'P and q values in RNA Seq' and find the detailed difference. Anyway, the DEGs need to be determined by q value rather than p value.

Response 2

The DEGs are now presented following analysis using q value based on false discovery rate (FDR) adjusted p values using the Benjamini and Hochberg method. For example, a revised Fig. 3a – see below, is now presented with the data plotted using FDR. The DEGs chosen with a FDR value cut off of < 0.05 are displayed in Fig. 3b.

In addition, we have performed qPCR on selected transcripts for further validation, including eight of the lipid genes, three transcription factor genes, and four ROS response genes (Fig. 3c, e, & f – see above, and in Supplementary Fig. 8 – see below). This data is described in lines 227-230, 236-244, 259-263, and 281-288.

Revised Fig. 3a:

Figure 3 (a) 'Volcano' plot of differentially expressed genes in cells 1 day after irradiation with 5 Gy at 0.5 Gy/min relative to control (non-irradiated) cells. Highlighted genes were significantly up-regulated (266 genes) or down-regulated (144 genes) following irradiation.

New Supplementary Fig. 8:

Supplementary Figure 8. Relative expression of selected genes related to lipid metabolism, transcriptional regulation and reactive oxygen species responses in response to irradiation and compared to control (non-irradiated) treatment, as determined by qPCR. Expression levels were determined relative to 18S expression. All data are means \pm standard error ($n = 3$) and the differences compared to control are all significant at $P < 0.05$.

Comment 3	Response 3
2 Line 500 'Quality and integrity of the RNA samples were assessed using a 2200 TapeStation' Please specify the quality assessment results of RNA samples, e.g., RIN number.	RIN number values are now provided in the Methods text (lines 565-567). The samples had RIN values ranging from 7.1 to 9.4. All samples were deemed to be sufficient for generation of cDNA libraries.
Comment 4	Response 4
3 Line 548: 'Differences between treated samples and controls were tested using a non-parametric two-tailed Mann–Whitney U test'. In statistical analysis, a normality test is usually performed, where data passed it an one-way ANOVA is applied, otherwise non-parametric analysis is used. Please justify why non-parametric test was applied for all results without normality test.	For small sample sizes, normality tests have little power to reject the null hypothesis and therefore small samples most often pass normality tests (10.5812/ijem.3505; 10.1111/1467-9884.00122). We had from 3 to over 25 measurements, depending on the method of analysis. The majority, if not all, of these sample sizes are small from the point of general statistics.

REVIEWERS' COMMENTS:

Reviewer #1 (Remarks to the Author):

The purpose of the present work was to analyze the effect of low-doses (1–5 Gy) of ionizing radiation in the X-ray range, on *Chlorella sorokiniana* (a green microalga) in terms of biological and biochemical responses. Authors have established that low-doses of ionizing radiation can induce a non-toxic and rapid (24hs) hormetic response, promoting the upregulation of > 30 lipid metabolism genes, such as genes encoding an acetyl-CoA carboxylase subunit, phosphatidic acid phosphatase, lysophosphatidic acid acyltransferase, and diacylglycerol acyltransferase. The paper is well written and designed. In addition, the experimental work is original and results interesting. Authors have performed a detailed response covering all the aspects pointed by this reviewer in the revision process. Thus, the manuscript deserves publication in *Communications Biology*.

Reviewer #2 (Remarks to the Author):

After a second review, the manuscript is still well-written and I believe will make meaningful contributions to the field of algal bioproduct production.

The authors have addressed the reviewer's comments and by doing so have improved the statistical rigor and streamlined the results/ discussion of this article. I have no further comments other than support for this paper.

Reviewer #3 (Remarks to the Author):

All the concerns have been addressed and the manuscript can be accepted in the current form.